# Levodopa ONOFF-state freezing of gait: Defining the gait and non-motor phenotype

**Reid D. Landes[1], Aliyah Glover[2], Lakshmi Pillai[2], Shannon Doerhoff[2], Tuhin Virmani** **[2,3]***

**1** Department of Biostatistics, University of Arkansas for Medical Sciences, Little Rock, AR, United States of America, **2** Department of Neurology, University of Arkansas for Medical Sciences, Little Rock, AR, United States of America, **3** Center for Translational Neuroscience, University of Arkansas for Medical Sciences, Little Rock, AR, United States of America

* TVirmani@uams.edu

## Abstract

### Background

Freezing in the levodopa-medicated-state (ON-state) is a debilitating feature of Parkinson's disease without treatment options. Studies detailing the distinguishing features between people with freezing of gait that improves with levodopa and those whose freezing continues even on levodopa are lacking.

### Objective

To characterize the gross motor, gait, and non-motor features of this phenotype.

### Methods

Instrumented continuous gait was collected in the levodopa-medicated-state in 105 patients: 43 non-freezers (no-FOG), 36 with freezing only OFF-levodopa (OFF-FOG) and 26 with freezing both ON- and OFF-levodopa (ONOFF-FOG). Evaluation of motor and non-motor disease features was undertaken using validated scales. A linear mixed model with age, sex, disease duration, and motor UPDRS scores as covariates was used to determine differences in spatiotemporal gait and non-motor disease features among the groups.

### Results

Compared to OFF-FOG, the ONOFF-FOG group had greater disease severity (on the Unified Parkinson's disease Rating Scale) and worse cognition (on the Montreal Cognitive Assessment, Frontal Assessment Battery and Scales for Outcome in Parkinson's disease-Cognition scales) and quality of life (on the PDQ-39), but similar mood (on the Hamilton depression and anxiety scales) and sleep quality (on Epworth sleepiness scale and RBD questionnaire). For several gait features, differences between the ONOFF-OFF groups were at least as large and in the opposite direction as differences between OFF-no groups, controlling for disease severity. Variability in ONOFF-FOG was greater than in other groups. Using results from our study and others, a power analysis for a potential future study reveals

**Data Availability Statement:** All relevant data are within the paper and its Supporting Information files.

**Funding:** This work was supported in part by the University of Arkansas Clinician Scientist Program

(TV), NIGMS P30 award (GM110702) (TV) and the Parkinson's Foundation (PF-JFA-1935) (https://www.parkinson.org/;TV). The funders did not play any role in the study design, data collection and analysis, decision to publish or preparation of the manuscript.

**Competing interests:** The authors have declared that no competing interests exist.

sample sizes of at least 80 ONOFF and 80 OFF-FOG patients would be needed to detect clinically meaningful differences.

## Conclusions

Intra-patient variability in spatiotemporal gait features was much greater in ONOFF-FOG than in the other two groups. Our results suggest that multifactorial deficits may lead to ONOFF-FOG development.

## Introduction

Freezing of gait (FOG), due to its impact on quality of life and fall risk, is one of the more debilitating motor features of Parkinson's disease (PD). The incidence of FOG ranges from 7% in early disease [1], and up to 92% near death [2]. Initially FOG is usually levodopa responsive (OFF-FOG), which may mask its presence until motor fluctuations develop and increased time in the motor OFF state leads to more bothersome freezing. There is also a population of PD patients that develop more resistant, levodopa unresponsive freezing, which has been termed levodopa-unresponsive freezing or ONOFF-FOG [2, 3] to suggest that freezing occurs in these patients in both the ON and OFF levodopa states, irrespective of levodopa dosing. In a previously published autopsy cohort [2], the group that developed ONOFF-FOG did so approximately 10 years after they developed OFF-FOG [2, 4]. Patients with ONOFF-FOG sub-jectively report little to no benefit in freezing episodes with their levodopa doses, and often have had worsening FOG with trials of increased levodopa dosing [5]. People with ONOFF-FOG also can simultaneously experience dyskinesias, while having freezing episodes, suggest-ing an otherwise optimally medicated ON-state [6]. A third group of patients who do not experience FOG in the OFF state, but only the ON-state has also been suggested [7] but this phenomenon is not common [3, 6].

It is unclear why only some PD patients develop ONOFF-FOG, and given the lack of dopa-minergic response, whether ONOFF-FOG is just a continuum to a more severe freezing state, or a different phenomenon all together. We could find no studies that have reported on the comparison between gait or non-motor phenotypes of OFF- and ONOFF-FOG subtypes to help elucidate differences between the two. However, a few groups have explored select fea-tures of ONOFF-FOG compared to either no-FOG, OFF-FOG or a combination thereof and therefore results cannot be combined and are summarized individually.

Lucas McKay et al. [6] performed a levodopa challenge in PD participants, with 45/55 show-ing a >20% levodopa improvement on the movement disorders society Unified Parkinson's Disease Rating scale (MDS-UPDRS) motor scores. Of these 76% had dyskinesias and 19 were classified as ONOFF-FOG, 11 OFF-FOG and 15 non-freezers (no-FOG) based on the UPDRS FOG item scores in the OFF and ON state. ONOFF-FOG patients had a range of UPDRS scores that spanned the other groups, and had a more severe phenotype than OFF-FOG patients based on the new freezing of gait questionnaire (N-FOGQ) and MDS-UPDRS activities of daily living (ADL) scores. There was no variation in levodopa response between groups in this study. In an earlier study from the same group, Factor et al. [8] showed that ONOFF-FOG patients had older age and worse UPDRS motor scores than no-FOG and worse visuospatial and executive function than both no-FOG and OFF-FOG. Ferraye et al. [9] used retrospective data available on PD patients who had undergone DBS placement and classified freezers and ONOFF-FOG (levodopa resistant or L-FOG in their manuscript) based on persistent scores on UPDRS ADL

FOG Item scores in the ON-levodopa state. They found that ONOFF-FOG patients had worse executive function compared to other PD patients that were a combined group of OFF-FOG and no-FOG. Moretti et al. [10] also classified freezers based on the UPDRS ADL FOG Item score and found that compared to OFF-FOG, ONOFF-FOG patients did worse on a digit span forward task, made more mistakes on the Trail making test and Proverb's interpretation task, and had a longer execution time and made greater mistakes on the STROOP test and the ten point clock test. Using the Cornell evaluation for depression they suggested that OFF-FOG patients were more depressed than ONOFF-FOG, but they found no differences in the anxiety sub-score of this scale between the groups. In the DEEP-FOG study by Amboni et al. [3], FOG patients (based on item 3 of the FOG-Q) were classified subjectively as ON-FOG, OFF-FOG or ONOFF-FOG based on questions asking them whether they froze in the best (ON), worst (OFF) or both (ONOFF). Of 593 patients, 6 were classified as ON-FOG, 200 OFF-FOG and 119 ONOFF-FOG. Compared to OFF-FOG, ONOFF-FOG patients had higher motor-UPDRS scores, were older at motor onset, were on lower levodopa doses, had lower MMSE scores, and reported similar quality of life on the Parkinson's disease questionnaire-8 (PDQ-8).

The goal of our study was therefore to identify an objectively quantifiable feature set from gait, gross-motor and non-motor assessments that distinguish between people with PD that have OFF-FOG, ONOFF-FOG or no-FOG phenotypes. A better understanding of these different phenotypes is essential in order for targeted therapeutic options to be developed.

## Materials and methods

### Protocol approvals, registrations, and patient consents

Patients were recruited from the Movement Disorders Clinic at the University of Arkansas for Medical Sciences (UAMS) from September 2014 to August 2019. The study was approved by the local Institutional Review Board (UAMS IRB# 203234), written informed consent was obtained from all patients, and the study was conducted in accordance with the guidelines of the Declaration of Helsinki.

### Study population

Patients with PD based on UK brain bank criteria were enrolled. People with a very high fall frequency of greater than 1 fall per day (due to safety concerns related to objective gait assessments), severe cognitive dysfunction with a Montreal Cognitive Assessment (MoCA) [11] score <10, or those with anti-dopaminergic medication use in the prior year were not enrolled. All patients were evaluated in the levodopa medicated or ON-levodopa state. For the study, patients were allowed to take their regular medication doses, including levodopa doses, as per their normal dosing schedule. The dose and time from last levodopa dose were documented during the study visit and reported in Table 2.

Patients with a score of 0 on Item 3 of the Freezing of Gait Questionnaire (FOG-Q) [12] were classified as non-freezers (no-FOG). Patients with a score of 1 or more on Item 3 of the FOG-Q, and/or had evidence of FOG on examination by a movement disorders trained neurologist (T. V.) were categorized as freezers (FOG). FOG patients were characterized as OFF-freezers if they 1) reported subjective resolution of freezing with levodopa despite improvement in other motor symptoms such as tremor, rigidity or bradykinesia, and 2) were not witnessed to freeze during their study evaluations that were performed in the levodopa medicated or ON-levodopa state (by T.V). FOG patients were categorized as ONOFF-FOG if they had freezing of gait visualized on examination during study assessments, which were performed in the ON-levodopa state, including the UPDRS and gait assessments (by T.V.), and met at least one of the following 3 criteria: 1) they reported no improvement or worsened freezing after their current daily levodopa doses (for

example gait freezing was better in the morning prior to their first levodopa dose in the morning, and worsened after they took the morning dose), 2) they reported lack of improvement in freezing, or worsened freezing with clinical levodopa dose increase trials documented in the UAMS Movement disorders clinic medical record, or 3) they reported freezing while also dyskinetic suggesting a dopamine saturated state. All FOG patients were clinically treated in the UAMS Movement disorders clinic (by T.V.) and therefore had multiple assessments prior to study enrollment and results from prior levodopa adjustments contributed to the accuracy of the categorization. No patients reported freezing in the ON-levodopa only and not the OFF-levodopa state, previously defined as ON-FOG [7]. Of the 105 PD subjects enrolled, 43 were categorized as no-FOG, 36 as OFF-FOG, and 26 as ONOFF-FOG. The differences between measured gross motor, gait and non-motor features between the noFOG to OFF-FOG were used to compare and contrast the differences between the OFF-FOG and ONOFF-FOG groups.

## Gait kinematics

Patients walked at a "comfortable" pace, 8 lengths of a 20 foot × 4 foot instrumented gait mat, in the levodopa medicated or ON-levodopa state, and data were collected and analyzed using Protokinetics Movement Analysis Software (PKMAS, Protokinetics, Havertown, PA). Freezing episodes were excluded from analysis during manual foot-print review (by AG and LP). The mean and coefficient of variability (CV) for steady state gait was calculated for 10 spatiotemporal features of gait; stride-length, stride-width, stride-time, stride-velocity, stance-phase-percent, total-double-support-percent, integrated-pressure applied during a step, foot-strike-length, cadence, and ambulation time. The definitions of these gait features are provided in Table 1. The more affected side was determined based on the ratio of the summated right/left scores for UPDRS items 20–26. Gait asymmetry was calculated as the ratio of the more/less affected side for each gait feature.

## Other assessments

The non-gait assessments were also conducted in the levodopa ON-state and included the Unified Parkinson's Disease Rating Scale (UPDRS) [13], the FOG-Q [12], the MoCA [11], the

**Table 1. Spatiotemporal gait feature definitions.**

| Measure | Definition |
|---|---|
| Stride-length (cm) | the distance between heel strikes of two consecutive footsteps of the same foot, i.e., two right or two left heel strikes along the direction of travel (x-axis) |
| Stride-width (cm) | the distance between heel strikes of two consecutive footsteps of the same foot, i.e., two right or two left heel strikes, perpendicular to the direction of travel (y-axis) |
| Stride-time (s) also known as gait cycle time | the time difference (s) between the initial heel contacts with the mat of two consecutive footsteps of the same foot, i.e., two right or two left. |
| Stride-velocity (cm/s) | the stride length divided by stride time, calculated for each gait cycle. |
| Stance-phase-percent | percentage measure of time spent in stance phase of the gait cycle and calculated as stance time/gait cycle time. |
| Total-Double-Support-phase-percent | percentage measure of time spent in the double support phase of the gait cycle and calculated as total double support time/gait cycle time. |
| Integrated-pressure (pressure x s) | the sum of pressure applied by a footstep at each sampling time (120 Hz sampling rate) in the area of its contact with the ground. |
| Foot-strike-length (cm) | length of the major axis of the ellipse enclosing each footstep; PKMAS creates an ellipse around each footstep during footstep identification. |
| Cadence (steps/minute) | total footsteps / cumulative time (in minutes) to walk 8 lengths of the mat |
| Ambulation time (s) | total time taken to walk 8 lengths of the mat. |

Frontal Assessment Battery (FAB) [14], the Scales for Outcome in Parkinson's disease–Cognition (SCOPA-Cog) [15], the Hamilton depression (HAM-D) [16] and anxiety (HAM-A) [17] rating scales, Apathy Evaluation Scale (AES) [18], Parkinson's disease questionnaire for quality of life (PDQ-39) [19], Epworth Sleepiness scale (ESS) [20], and REM Sleep Behavior Disorder Screening Questionnaire (RBD-Q) [21]. Levodopa equivalent daily dose was calculated using previously accepted formulas for conversion of levodopa extended release formulations and dopamine agonist doses [22, 23].

## Statistical analysis

Summary statistics (means ± standard deviations, percentages) of patient characteristics for the three FOG groups are provided in Table 2. For non-gait features, we used a linear mixed model that accounted for FOG group, sex, age, and disease duration; we also allowed each FOG group to have its own variance, so as not to bias the standard errors and increase the chance of rejecting a true null hypothesis. Our primary comparison of interest within each analysis was between the OFF- and ONOFF-FOG groups; and to corroborate known differences between OFF- and no-FOG groups, we also compared these two groups. These comparisons were made with contrasts within the linear mixed model. For gait features, we added motor UPDRS as a covariate and the interaction of FOG group with motor UPDRS to the mixed model. When the interaction between FOG group and motor UPDRS was significant, we made comparisons at low (9.5), middle (13.0), and high (16.0) values of motor UPDRS; specifically, the lower value was

**Table 2. Demographics and motor metrics of patients.**

|  |  | no-FOG (n = 43) | OFF-FOG (n = 36) | ONOFF-FOG (n = 26) |
|---|---|---|---|---|
| General features: |  |  |  |  |
|  | Female | 42% | 50% | 38% |
|  | Right-handed | 91% | 92% | 96% |
|  | Right side more affected | 56% | 56% | 50% |
|  | PIGD phenotype at visit | 40% | 78% | 100% |
| Initial symptom: | rest tremor | 53% | 31% | 38% |
|  | gait | 14% | 22% | 35% |
|  | Age at motor onset (years) | 60.9 ± 9.6 | 55.6 ± 10.9 | 60.2 ± 9.7 |
|  | Disease duration (years) | 6.2 ± 4.9 | 9.4 ± 6.4 | 10.4 ± 5.1 |
|  | Age at enrollment (years) | 67.1 ± 8.2 | 64.9 ± 8.6 | 70.5 ± 8.1 |
|  | Hoehn & Yahr score | 1.8 ± 0.5 | 2.2 ± 0.5 | 3.2 ± 0.9 |
|  | Fall frequency (per month) | 0.1 ± 0.2 | 1.1 ± 5.0 | 8.1 ± 18.4 |
|  | FOG duration (years) | - | 2.2 ± 2.4 | 3.4 ± 2.1 |
| Medications: |  |  |  |  |
|  | Daily levodopa dose (mg/day) | 533 ± 308 | 717 ± 411 | 989 ± 406 |
|  | levodopa per dose (mg/dose) | 150 ± 64 | 193 ± 121 | 235 ± 99 |
|  | Time from levodopa dose (hrs) | 2.7 ± 2.3 | 2.0 ± 1.1 | 1.2 ± 0.6 |
|  | Duration on levodopa (years) | 2.8 ± 2.9 | 5.6 ± 5.8 | 7.7 ± 4.8 |
|  | On dopamine agonist at visit | 19% | 22% | 15% |
|  | On MAO-I at visit | 40% | 25% | 15% |
|  | LEDD (l-dopa+agonist+MAO-I) | 588 ± 324 | 786 ± 413 | 1032 ± 402 |

Values for continuous variables are mean ± standard deviation, and for categorical variables are percent of the relevant group sample size (n). PIGD: Postural Instability Gait Disorder variant; FOG-Q: Freezing of Gait Questionaire; FOG: Freezing of Gait; MAO-I: monoamine oxidase inhibitor; l-dopa: levodopa; LEDD: levodopa equivalent daily dose.

the 25<sup>th</sup> percentile for the OFF-FOG group, 13.0 was the median among no-FOG and OFF-FOG patients, and 16.0 was the 75<sup>th</sup> percentile for the no-FOG group. For all but 3 features, however, the interaction had $p>0.10$; for those features, we dropped the interaction term from the model, and compared OFF-FOG to ONOFF-FOG. We note these OFF-FOG vs ONOFF-FOG comparisons account for motor UPDRS so that the two groups are compared at the same motor UPDRS values. Figs 2 and 3 and S1 show the differences along with their 95% confidence intervals for those differences for all the features we analyzed. In order to concisely visualize these differences from 8 to 10 features within a measure type, all having different scales of measurement, we standardized the differences (see the S1 Table with original and standardized numerical results). The S1 File contains a thorough description of the statistical models used.

One patient, #96, had extreme values for most features in all the measure types; and another patient, #80, had extreme values for CV measures. We excluded patient #96 from all analyses, and patient #80 from analyses of CVs. However, to learn whether and how inferences might change with these patients included, we re-ran the above analyses again, and present the results in S1 Fig. The differences in inferences from the reduced dataset to the full dataset were minor: the full dataset revealed significant differences in mean stance phase percent and total double-support percent between OFF-FOG and ONOFF-FOG groups.

## Results

Demographics of the PD patients enrolled are shown in Table 2. We did not statistically compare these characteristics among the three groups as we had no hypotheses necessitating such tests. Ages at motor onset and duration of disease did not differ by more than a standard deviation (SD) among FOG groups. Consequently, ages at enrollment also did not differ by more than a standard deviation. Regarding initial symptoms at motor onset, though the ONOFF-FOG group had the highest percentage of patients whose first symptom was in gait, over a third of ONOFF-FOG patients reported rest tremor as their initial symptoms. Hoehn & Yahr scores in the ONOFF-FOG group were higher than the other two groups by about 1½ standard deviations or more and fall frequencies in the ONOFF-FOG group were over 7 times that in the other two groups.

Controlling for age, disease duration, and sex, ONOFF-FOG patients had statistically greater disease severity with higher scores on both the motor-related and non-motor related questions of the UPDRS (Table 3) than in OFF-FOG; consequently, the total UPDRS score was also statistically higher in ONOFF-FOG. Cognitive scores were statistically lower on the MoCA, FAB, and SCOPA-Cog in ONOFF-FOG patients compared to OFF-FOG patients. Quality of life (as measured by the PDQ-39) was also statistically worse in ONOFF-FOG patients. While apathy scores on the AES were greater in OFF-FOG patients by 3.4 points, no difference (i.e. a difference of 0) was also plausible. Finally, there was little to no evidence that these two FOG groups differed in depression (HAM-D), anxiety (HAM-A), or sleep quality (RBD and Epworth Sleepiness Questionnaires).

Based on individual question responses on the FOG-Q, mean frequency and duration were both about 1 standard deviation higher in ONOFF-FOG patients than the means in their OFF-FOG peers (3.31±0.62 vs 2.00±1.12 score for item 3 frequency, 2.31±0.93 vs 1.47±0.97 item 4 longest duration and 1.88±0.75 vs 1.06±0.80 for average of items 5–6, average duration).

### Gait kinematics

We controlled for sex, age, disease duration, and motor UPDRS score in all analyses of spatio-temporal gait features. For all but three features, there was little to no evidence that the slope of motor UPDRS depended on FOG group (all interaction $p$s $\geq$ 0.130). Compared to no-FOG

patients, OFF-FOG patients had statistically greater total double support percent (see Fig 1A for example), stance phase percent, ambulation time, and stride width, slower stride velocity, and shorter stride length; the CV was statistically greater in stance phase percent and foot-strike length (Fig 2A and 2B, black circles). Importantly, the estimated differences between ONOFF-FOG and OFF-FOG patients were *at least as large*, and in the opposite direction, as the statistically significant differences between no-FOG and OFF-FOG patients for mean total double support percent and stance phase percent, and CV of foot-strike length. However, due to high variability in ONOFF-FOG patients (SDs of these features were 2.1 to 2.5 times greater in the ONOFF-FOG patients), these differences were not statistically significant for the OFF--ONOFF comparison (Fig 1A for example, and Fig 2A and 2B, gray squares). For estimated mean differences, CVs of stride time, integrated pressure, and total double support percent were also at least as large, and in the opposite direction between OFF vs ONOFF-FOG compared to OFF vs noFOG comparisons, but none of these comparisons were statistically significant. The importance of these results are also reviewed in the discussion section and used to provide sample sizes for future studies.

For three features, there was evidence that the slope of motor UPDRS depended on FOG group (all interaction $ps \leq 0.089$). As motor UPDRS increased, CV of stride-length and stride-velocity increased significantly faster in OFF- and ONOFF-FOG patients than in no-FOG patients where motor UPDRS had a near-0 slope (see Fig 1B for example). At midrange and higher values of motor UPDRS, OFF-FOG patients had significantly more variability in stride-length and stride-velocity than in the no-FOG patients; the CVs were between 0.9 and 1.75 percentage points (*pp*) higher in the OFF-FOG patients. We note that CV in ONOFF-FOG patients was (a constant) 1.5 *pp* higher than OFF-FOG patients for stride-length and (a

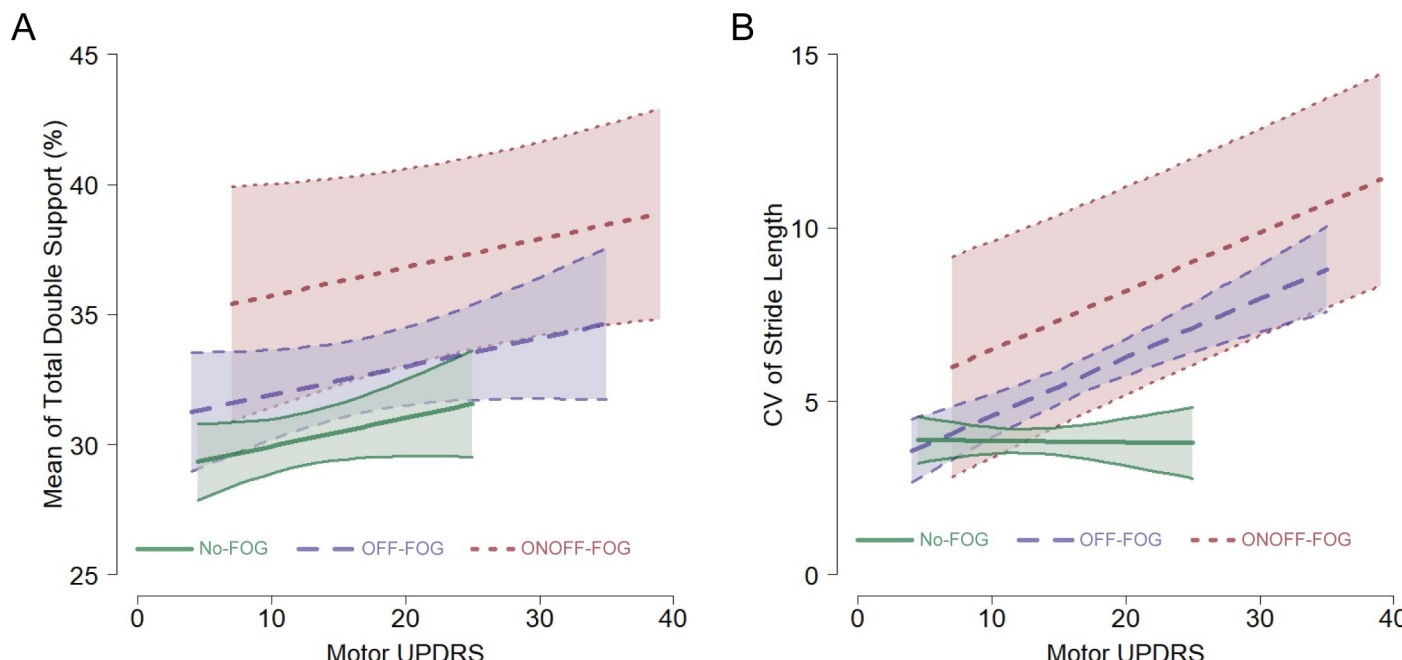

**Fig 1. Representative examples of gait features as a function of motor UPDRS. (A)** Slope independent of FOG group. Mean total double support percent is plotted as a function of motor UPDRS scores with mean and 95% confidence intervals shown, as an example of a gait feature where the slope of UPDRS on the feature did not depend upon FOG group; i.e., was the same among the three FOG groups. **(B)** Slope dependent upon FOG group. CV stride length is plotted as a function of motor UPDRS scores with mean and 95% confidence intervals shown, as an example of a gait feature where the slope of UPDRS on the feature depended upon FOG group; i.e., the slopes were not the same for all FOG groups. Note: The estimated regressions lines for each FOG group span the observed range of UPDRS scores observed for that group.

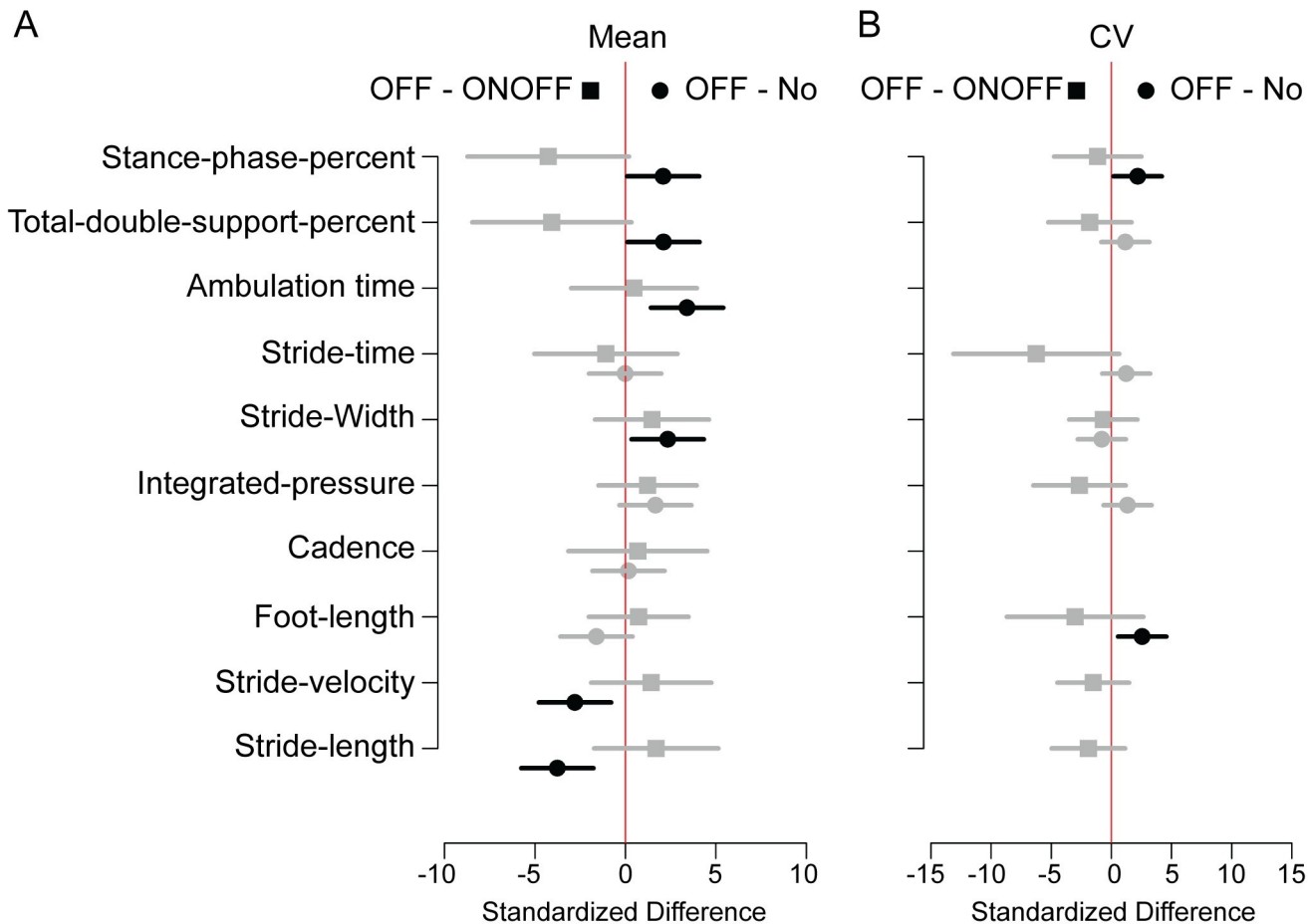

**Fig 2. Comparison of spatiotemporal features for OFF-state freezers compared to ONOFF-state freezers and non-freezers.** The standardized difference between OFF-state freezers and ONOFF-state freezers (OFF-ONOFF, squares) and OFF-state freezers and non-freezers (OFF-No, circles) is shown for **(A)** mean and **(B)** stride-to-stride variability (coefficient of variation, CV). Bars denote the 95% confidence intervals. Comparisons with significant group differences are represented by black symbols and bars, while those without significant group differences are represented with gray symbols and bars.

constant) 1.9 *pp* higher for stride-velocity, but as the residual standard deviation of ONOFF--FOG patients was at least 3.6 times greater than that for the OFF-FOG patients, these differences were not significant.

## Gait asymmetry

To determine if asymmetry in gait played a role in developing ONOFF-FOG, we analyzed the ratio of each spatiotemporal gait feature for the more affected/less affected side for each patient. We found no statistical differences when comparing OFF-FOG patients to each of ONOFF-FOG and no-FOG patients for either mean (Fig 3A) or CV (Fig 3B). However, comparisons of CV ratios between OFF- and ONOFF-FOG were more extreme and in the opposite direction as the same comparisons between OFF-and no-FOG for foot length, stride time, stride velocity, stance phase percent, and total double support percent (Fig 3B).

## Multiple testing

In Figs 2 and 3 and S1 and Table 3, we conducted 80 comparisons. Of those, 18 were significant at the 0.05 significance level. We computed the positive False Discovery Rate, which

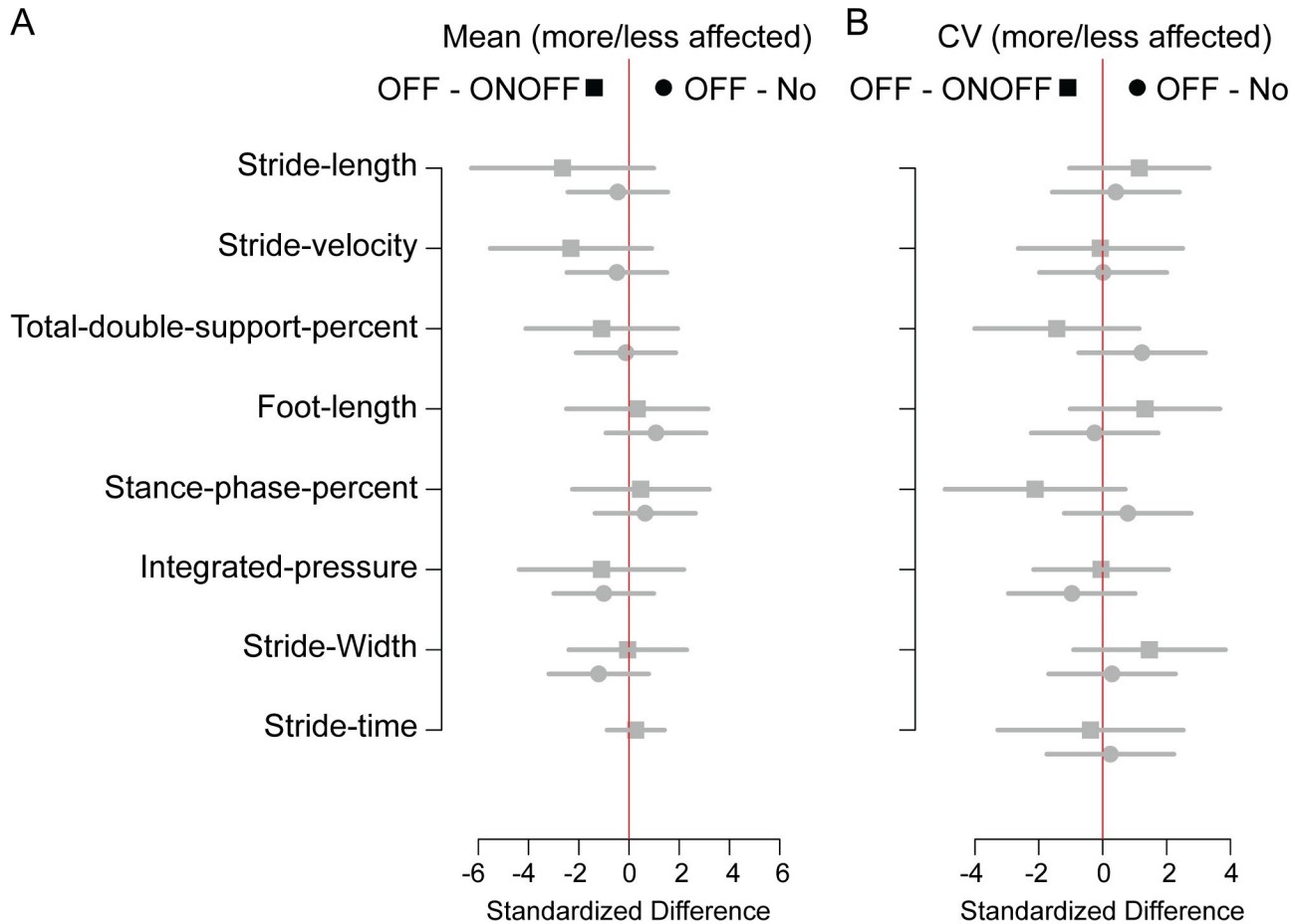

**Fig 3. Comparison of asymmetry between more and less affected side for spatiotemporal parameters in OFF-state freezers compared to ON-state freezers and non-freezers.** The standardized differences in the asymmetry ratio (more/less affected side) are also shown for **(A)** mean and **(B)** CV. Bars denote the 95% confidence intervals. Comparisons with significant group differences are represented by black symbols and bars, while those without significant group differences are represented with gray symbols and bars.

estimates how many of the significant results may be Type I errors [24]. The positive False Discovery Rate was 0.14, with a 95% confidence upper bound of 0.23; this means, we are 95% confident that there were less than 5 false discoveries in the 18.

## Discussion

In this manuscript we report the motor, non-motor, and gait phenotypes of PD patients with freezing of gait that occurs in the levodopa ON- and OFF-state (or levodopa unresponsive freezing) compared to those with freezing that occurs only in the levodopa OFF-state (or levodopa-responsive freezing), using objective testing. In our cohort of FOG patients with approximately the same disease duration, and age at onset, ONOFF-FOG patients had greater cognitive dysfunction [3, 8, 10], greater motor disease severity [3, 8], and lower quality of life [6] than OFF-FOG patients. Regarding spatiotemporal gait, in several features, the separation in means between OFF-FOG and ONOFF-FOG patients was at least as large, and in the opposite direction, as the separation in means between OFF-FOG and no-FOG patients; this, accounting for sex, age at onset, disease duration, and disease severity (motor UPDRS score). The variability of these spatiotemporal gait features in ONOFF-FOG patients was also

**Table 3. Means (and 95% CIs) for motor and non-motor features, adjusted for sex, age at enrollment, and disease duration.** The differences in means (and 95% CIs) between the OFF and ONOFF-FOG groups are also provided, with the test statistics and *p*-values. The CIs for differences in means that do not contain 0 are in bold font.

| Outcome | no-FOG (n = 43) | OFF-FOG (n = 36) | ONOFF-FOG (n = 26) | OFF-FOG–ONOFF-FOG Estimate (95% CI) | *t*-statistic[a] *p*-value |
|---|---|---|---|---|---|
| Total UPDRS Score | 21.9 | 31.7 | 46.4 | **-14.7** | -5.32 |
| | (18.7, 25.1) | (28.2, 35.2) | (42.2, 50.6) | **(-20.2, -9.2)** | < .001 |
| UPDRS Part III (motor) | 12.3 | 16.8 | 25.7 | **-8.9** | -4.81 |
| Score | (10.1, 14.5) | (14.5, 19.2) | (22.9, 28.5) | **(-12.5, -5.2)** | < .001 |
| UPDRS Part I+II (non-motor) | 9.6 | 14.8 | 20.7 | **-5.8** | -3.99 |
| Score | (7.9, 11.3) | (13.0, 16.7) | (18.5, 22.9) | **(-8.8, -2.9)** | < .001 |
| FOG-Q score | 2.0 | 8.3 | 14.1 | **-5.8** | -6.79 |
| | (1.0, 3.0) | (7.2, 9.4) | (12.8, 15.4) | **(-7.5, -4.1)** | < .001 |
| FOG-Q sub-score q3-6 | 0 | 5.4 | 9.5 | **-4.1** | -6.60 |
| (freezing) | (-0.7, 0.7) | (4.6, 6.2) | (8.6, 10.5) | **(-5.4, -2.9)** | < .001 |
| FOG-Q sub-score q1-2 | 2.0 | 2.9 | 4.5 | **-1.6** | -4.23 |
| (non-freezing) | (1.6, 2.5) | (2.4, 3.4) | (4.0, 5.1) | **(-2.4, -0.9)** | < .001 |
| MoCA score | 25.8 | 25.3 | 21.8 | **3.4** | 3.77 |
| | (24.7, 26.8) | (24.1, 26.4) | (20.5, 23.2) | **(1.6, 5.2)** | < .001 |
| FAB | 16.1 | 15.0 | 13.5 | **1.5** | 2.43 |
| | (15.3, 16.8) | (14.2, 15.8) | (12.5, 14.4) | **(0.3, 2.8)** | .017 |
| SCOPA-Cog | 25.4 | 22.6 | 19.3 | **3.2** | 2.58 |
| | (23.9, 26.8) | (21.0, 24.2) | (17.4, 21.2) | **(0.7, 5.7)** | .012 |
| PDQ-39 | 27.5 | 43.6 | 58.5 | **-15.0** | -2.86 |
| Questionnaire | (21.4, 33.6) | (36.9, 50.2) | (50.6, 66.4) | **(-25.3, -4.6)** | .005 |
| Apathy | 9.0 | 16.4 | 13.0 | 3.4 | 1.73 |
| Evaluation scale | (6.7, 11.3) | (13.9, 18.9) | (10.0, 16.0) | (-0.5, 7.3) | .088 |
| HAM-D | 7.0 | 10.4 | 9.7 | 0.7 | 0.50 |
| | (5.4, 8.6) | (8.6, 12.1) | (7.6, 11.8) | (-2.1, 3.5) | .621 |
| HAM-A | 4.9 | 7.6 | 7.4 | 0.2 | 0.17 |
| | (3.6, 6.2) | (6.2, 9.0) | (5.8, 9.1) | (-2.0, 2.3) | .862 |
| RBD | 5.0 | 5.9 | 5.2 | 0.8 | 0.91 |
| Questionnaire | (4.0, 5.9) | (4.9, 6.9) | (3.9, 6.4) | (-0.9, 2.4) | .363 |
| Epworth Sleepiness | 7.6 | 7.8 | 9.1 | -1.3 | -1.13 |
| Questionnaire | (6.2, 9.0) | (6.3, 9.3) | (7.4, 10.9) | (-3.6, 1.0) | .259 |

UPDRS: Unified Parkinson's Disease Rating Scale; FOG-Q: Freezing of Gait Questionnaire; MoCA: Montreal Cognitive Assessment Score; FAB: Frontal Assessment Battery; SCOPA-Cog: Scales for Outcome in Parkinson's Disease-Cognition; HAM-D: Hamilton Depression scale; HAM-A: Hamilton Anxiety scale; RBD: REM sleep behavior disorder.

[a] All *t*-statistics had 99 degrees of freedom.

markedly greater than in OFF-FOG and no-FOG patients, with standard deviations being about twice as large. Consequently, comparisons between OFF- and ONOFF-FOG patients are not known as precisely as those between OFF- and no-FOG patients.

Because of the increased variability of spatiotemporal gait features in ONOFF-FOG patients, power was reduced in our study. We used summary statistics from our study and Amboni et al. Table 1 [3] to perform a power analysis for a potential future study. Fig 4 presents the power plotted by total sample size for several effect sizes, $\sigma$, where $\sigma$ is the standard deviation for the OFF-FOG population, after controlling for sex, age at onset, disease duration, and disease severity. The important assumption in the power analysis is that the standard deviation for the ONOFF-FOG group is twice that of the OFF-FOG group. Details of the power

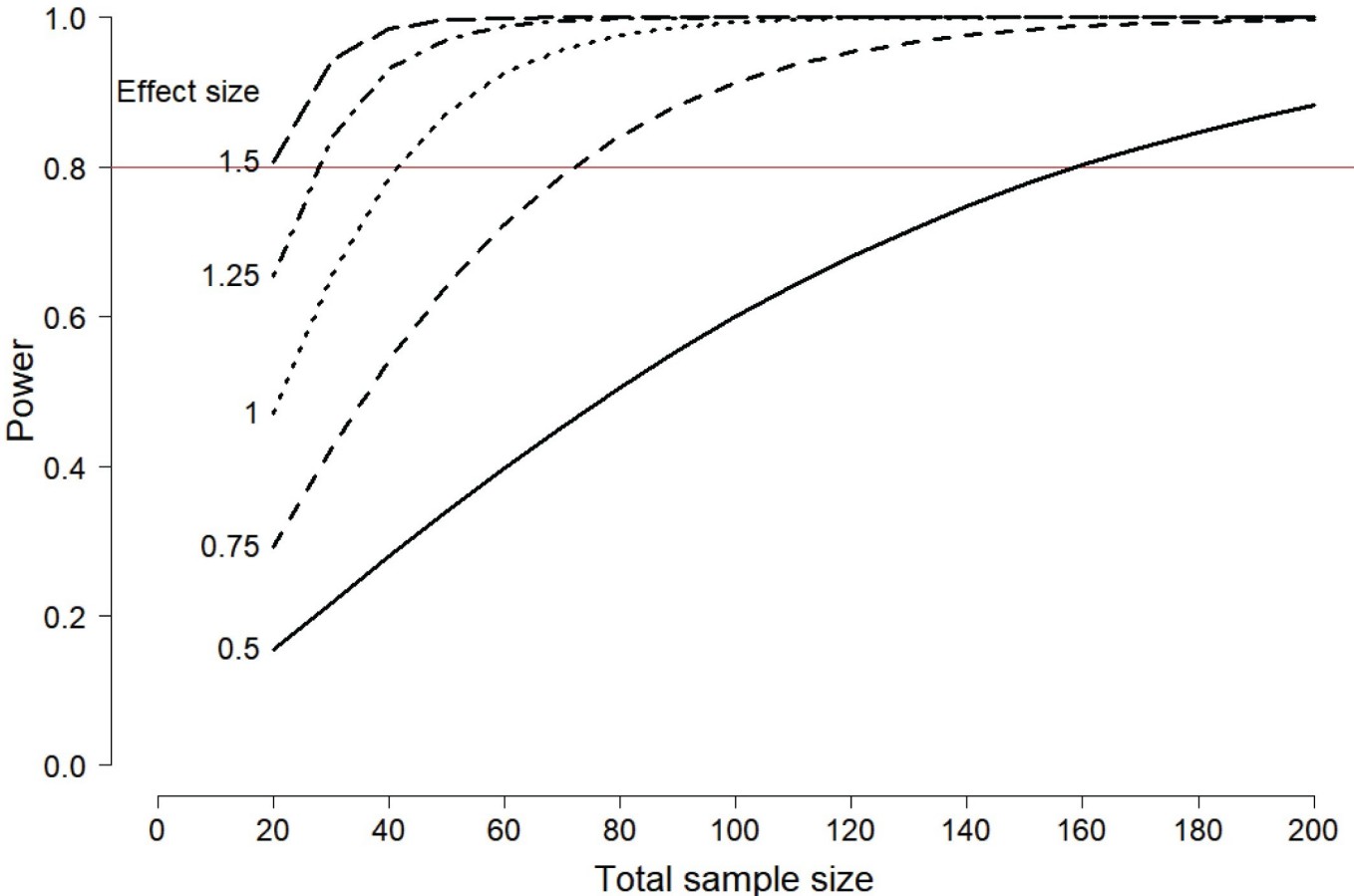

**Fig 4. Power plotted by total sample size for several effect sizes.** Effect sizes are expressed in terms of the population standard deviation, σ, of the OFF-FOG group, after accounting for sex, age at onset, disease duration, and disease severity. The power calculations assume that the population standard deviation of the ON-FOG group is 2σ. More details of the power analysis are in the S1 File.

analysis are in the S1 File. In order to detect a difference in means of size $\sigma/2$, 160 patients (80 OFF- and 80 ONOFF-FOG) would be needed to have 0.80 power on a 0.05 significance level test. We found statistical differences of size $\hat{\sigma}/2$ between OFF-FOG and noFOG groups in this study, but larger differences between OFF-FOG and ONOFF-FOG groups were not significant; see Fig 2A for example.

Our work corroborates prior work on several noted differences between those who have FOG while on levodopa (OFFON-FOG) and those who do not (OFF-FOG); specifically the differences were older age at onset [3, 8], worse UPDRS motor scores [3, 8], worse freezing severity [6], worse cognition scores (e.g., MMSE, MoCA or neuropsychological testing batteries) [3, 8, 10] and worse quality of life scores (e.g. UPDRS part II or PDQ) [6]. Worse quality of life, worse cognition and higher freezing severity scores were also reported in OFFON-FOG compared to a general PD group [9]. Importantly, we have gone beyond prior studies by examining objective gait measures, depression, anxiety, apathy, RBD and sleep quality on validated scales; to our knowledge, such evaluation between OFF and ONOFF-FOG groups have not been reported in previous studies.

Continuous gait abnormalities have been well documented in PD patients, and a recent meta-analysis showed consistent reduction in stride-length, swing-time and hip excursion compared to healthy controls [25]. In FOG patients continuous gait abnormalities including

decreased stride length setting [26–30], increased foot-strike length variability [28], and asymmetry in stride or steppage [31, 32] have suggested that the threshold to achieve motor program breakdown, and therefore a freeze is reduced in PD FOG patients [33]. In our study, mean stance phase and total double support, and the CV of foot-strike length in ONOFF-FOG patients were all greater than in OFF-FOG patients by margins that exceeded the confirmed increases found in the same features when comparing OFF-FOG to no-FOG patients (Fig 1). Because standard deviations of these features in ONOFF-FOG patients were over 2 times that in OFF-FOG patients, coupled with relatively small group sizes (26 ONOFF-FOG and 36 OFF-FOG patients), these results did not reach statistical significance. As a result, we cannot confidently say that these spatiotemporal gait dynamics differences reflect a fundamental difference in underlying disease pathology between the OFF-FOG and ONOFF-FOG groups. However, the fact that dopamine improves episodic freezing in one group and not the other, suggests that the pathways leading to break-down may involve more non-dopaminergic circuits in ONOFF-FOG.

PD-FOG patients have been reported to have greater sleepiness [34, 35] and REM sleep behavior disorder (RBD) [35, 36], than in no-FOG. The pedunculopontine nucleus (PPN) plays an important role in sleep-wake cycles and may play a role in integrating gait control with sleep function [37, 38]. PPN stimulation has also been proposed as a treatment for freezing of gait in Parkinson's disease [39, 40]. In our cohort, the ONOFF-FOG group were not significantly worse than those in the OFF-FOG group on sleep or RBD scales; further, 0 was in the middle half of the confidence intervals for the difference between the ONOFF- and OFF-FOG groups, suggesting that this pathway was not significantly different between the ONOFF-vs OFF-FOG phenotypes. Apathy [41], depression [35], and anxiety [35, 42], have also been reported to be worse in FOG patients compared to noFOG. Anxiety [43] and depression have also been reported to be possible predictors of future FOG development. In our cohort, 0 was in the middle third of the confidence intervals for the difference between ONOFF- and OFF-FOG groups for HAM-A and HAM-D scores, suggesting that limbic pathways may not lead to the differential pathophysiology.

In our cohort PD ONOFF-FOG patients had higher FOG-Q scores on questions 3–6, indicating a greater frequency and duration of freezing episodes than their OFF-FOG counterparts. While these questions are based on subjective report and can be confounded by recall bias, it would suggest that ONOFF-FOG patients not only have a greater deficit in gait initiation than OFF-FOG patients, but also a decreased ability to return to normal gait after freezing occurs. Future analysis of the freezing episodes themselves and the response of the freezing episodes to dopamine supplementation could help shed further light on this process. Additionally, anticipatory postural adjustments (APAs) have been previously reported to contribute to gait initiation deficits in PD patients [44] and FOG patients [45–47] and further work is needed to determine how these may defer between ONOFF- and OFF-FOG patients and whether impaired APAs impact the extent of freezing duration in the different groups.

While our study did not have an imaging component to provide us with pathophysiologic correlates, we can still speculate. It has been suggested that dysfunction in cholinergic pathways leads to FOG, with neocortical cholinergic denervation [48] and decreased vesicular acetylcholine transporter binding in the striatum, temporal and mesofrontal limbic regions [49]. Imaging during imagined gait has also suggested involvement of the cerebellar locomotor region [50]. It is possible that FOG is modulated by both dopamine and cholinergic pathways. Initial dopaminergic denervation leads to a dopamine responsive gait freezing (OFF-FOG) and with progression of disease, more cholinergic denervation leads to development of a dopamine insensitive freezing (ONOFF-FOG). No cases of ONOFF-FOG developing as an initial manifestation of freezing in PD have been reported to our knowledge. As cholinergic pathways

also modulate cognition, this could provide the link between freezing and cognitive decline. It is however still possible that ONOFF-FOG is a separate phenomenon altogether and further imaging and longitudinal progression studies monitoring multiple modalities of PD dysfunction would be needed to tease this out.

It should also be taken into consideration that freezing is likely on a continuum. People without reported freezing of gait, could be in the "honeymoon period" where their symptoms are better on levodopa and therefore they do not notice freezing develop till it progresses beyond a "micro freeze" that is undetectable visually on exam and does not affect function. Similarly freezing of gait that subjectively resolves with levodopa may also exist on a continuum with freezing that occurs irrespective of motor fluctuations in other levodopa sensitive symptoms such as rigidity and bradykinesia. Development of accurate objective freeze detection algorithms could help resolve these issues as well. We have also included people with freezing that did not improve with levodopa and people whose freezing worsened with their levodopa doses, or with higher levodopa doses in the same group and these could be different sub-phenotypes on the continuum as well.

In our cohort, there was an increasing percentage of people meeting the PIGD phenotype designation going from noFOG to OFF-FOG to ONOFF-FOG. It remains unclear whether these two phenotypes, PIGD and FOG are independent risk factors for one another, or possibly the same phenotype. Similar to the increasing incidence of FOG from 7% in early disease [1] to 92% at death [2], the PIGD phenotype has also been reported to increase with disease duration with one cohort study reporting 54% PIGD participants at initial visit, 73% at 4-year follow-up and 88% at 8-year follow-up [51], albeit limited by drop-out over this period in the total population due to deaths in the cohort. Disease features associated with the PIGD phenotype are also similar to FOG including longer disease duration, greater disease severity, lower cognitive scores, impaired postural adjustments and faster disease progression [51–54]. We previously showed that FOG participants show faster spatiotemporal gait decline than noFOG [55]. Clinically OFF-FOG can respond well to levodopa and when motor-fluctuations are not occurring in early PIGD, freezing may go unwitnessed either in the form of micro-freezes, or levodopa responsive freezing. Further work is needed in larger longitudinal cohorts to explore these two different possibilities. Whatever the case, exercise has been shown to help possibly with decreasing disease progression and helping acutely with improving gait, balance and FOG and should be consistently recommended to all PD patients [56–58].

There are limitations to our study. We did not explore the freezing episodes themselves, or the spatiotemporal features before and after freezing episodes such as the progressively shorter stride prior to entering a freeze or trembling of the legs while in the freeze. Future studies exploring the episodic aspects of the freezing may provide more physiologic insight to guide correlation with pathophysiologic dysfunction. Due to the more severe motor phenotype in ONOFF-FOG patients, we were unable to perform assessments in the levodopa OFF condition for this study, and therefore were unable to explore any differential dopamine responsiveness of the continuous gait phenomena between the FOG groups [5]. However, since the OFF-FOG group has improved freezing in the ON-levodopa state while the ONOFF-FOG group still has freezing in the ON-levodopa state, the differences in spatiotemporal gait features that remain in the ON-levodopa state are of primary interest.

In summary, despite similar disease duration to OFF-state freezers, ONOFF-state freezers had greater motor disease severity and cognitive deficits, and lower quality of life, but sleep quality, mood and apathy were similar. The gait phenotype suggests that differences seen between ONOFF- and OFF-FOG patients are at least as large as those seen between OFF- and no-FOG patients, but between-patient variability was larger among ONOFF-FOG patients. These findings suggest multifactorial deficits leading to an ONOFF-FOG phenotype.

## Supporting information

**S1 Fig. Comparison of spatiotemporal parameters for OFF-state freezers compared to ON-state freezers and non-freezers for all participants including those with extreme values.** The standardized difference between OFF-state freezers and ON-state freezers (OFF-ON, squares) and OFF-state freezers and non-freezers (OFF-No, circles) is shown for **(A)** mean and **(B)** stride-to-stride variability (CV). The standardized differences in the asymmetry ratio (more/less affected side) are also shown for **(C)** mean and **(D)** CV. Bars denote the 95% confidence intervals. Comparisons with significant group differences are represented by black symbols and bars, while those without significant group differences are represented with gray symbols and bars.
(TIF)

**S1 Table. Original and standardized results.**
(PDF)

**S1 Dataset. Minimal dataset.**
(CSV)

**S1 File. Supplementary methods.**
(DOCX)

**S2 File. SAS code.**
(SAS)

**S3 File. R code.**
(R)

## Acknowledgments

We appreciate the mentorship of Dr. Edgar Garcia-Rill and Dr. Linda Larson-Prior. We also greatly appreciate the commitment and dedication of our patients, without whom this work would not be possible.

## Author Contributions

**Conceptualization:** Tuhin Virmani.

**Data curation:** Aliyah Glover, Lakshmi Pillai, Shannon Doerhoff, Tuhin Virmani.

**Formal analysis:** Reid D. Landes, Aliyah Glover, Lakshmi Pillai, Tuhin Virmani.

**Funding acquisition:** Tuhin Virmani.

**Investigation:** Aliyah Glover, Lakshmi Pillai, Shannon Doerhoff, Tuhin Virmani.

**Methodology:** Reid D. Landes, Tuhin Virmani.

**Project administration:** Tuhin Virmani.

**Supervision:** Tuhin Virmani.

**Writing – original draft:** Reid D. Landes, Tuhin Virmani.

**Writing – review & editing:** Reid D. Landes, Aliyah Glover, Lakshmi Pillai, Shannon Doerhoff, Tuhin Virmani.

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
