## [Decision Letter · Decision Letter 0]

8 Jul 2021

PONE-D-21-16241

Levodopa ON-state Freezing of Gait: Defining the Gait and Non-motor Phenotype

PLOS ONE

Dear Dr. Virmani,

Thank you for submitting your manuscript to PLOS ONE. This is an interesting area that caused some difference of opinion between reviewers. 

After careful consideration, we feel that it has merit but does not fully meet PLOS ONE’s publication criteria as it currently stands. Therefore, we invite you to submit a revised version of the manuscript that addresses the points raised during the review process.

In particular, in a design that has used traditional hypothesis testing, statements discussing features of data that were not statistically significant should be significantly shortened or eliminated, as suggested by one of the reviewers. Because of the direct relevance to unresponsive FOG, please see whether this work from our center would be an appropriate reference: https://www.nature.com/articles/s41531-019-0099-z

We look forward to receiving your revised manuscript.

Kind regards,

J. Lucas McKay, Ph.D., M.S.C.R.

Academic Editor

PLOS ONE

Journal Requirements:

3. We note that you have included the phrase “data not provided” in your manuscript. Unfortunately, this does not meet our data sharing requirements. PLOS does not permit references to inaccessible data. We require that authors provide all relevant data within the paper, Supporting Information files, or in an acceptable, public repository. Please add a citation to support this phrase or upload the data that corresponds with these findings to a stable repository (such as Figshare or Dryad) and provide and URLs, DOIs, or accession numbers that may be used to access these data. Or, if the data are not a core part of the research being presented in your study, we ask that you remove the phrase that refers to these data.

4. Please include a caption for figure 3.

Reviewers' comments:

Reviewer's Responses to Questions

**Comments to the Author**

1. Is the manuscript technically sound, and do the data support the conclusions?

Reviewer #1: No

Reviewer #2: No

Reviewer #3: Yes

2. Has the statistical analysis been performed appropriately and rigorously? 

Reviewer #1: No

Reviewer #2: No

Reviewer #3: Yes

3. Have the authors made all data underlying the findings in their manuscript fully available?

Reviewer #1: No

Reviewer #2: Yes

Reviewer #3: Yes

4. Is the manuscript presented in an intelligible fashion and written in standard English?

Reviewer #1: Yes

Reviewer #2: Yes

Reviewer #3: Yes

5. Review Comments to the Author

Reviewer #1: This manuscript aims to characterize persons with Parkinson’s disease (PD) who experience freezing of gait (FoG) while on parkinsonian medications and contrast these characteristics against those of persons with PD who either only experience FoG when off of their medications or do not experience FoG at all. The authors report that persons with PD + FoG when medicated had worsened disease severity, cognition, quality of life, and global gait characteristics than participants in the other groups. I found the topic of the study to be interesting; however, the reporting of the results is challenging to follow given the qualitative nature with which the results are reported and interpreted. The introduction also lacks important background that could help the reader to better understand the motivations behind the study.

ABSTRACT

In the Objective statement, I recommend revising “motor” to “gross motor”, given that gait is a motor function but mentioned separately.

In the final sentence of the Results section, please revise “Variability” to “Gait variability” (or, ideally, a more specific gait variability metric – e.g., step length variability).

In the Conclusions statement, I recommend revising or omitting the second sentence. This manuscript provides primarily descriptive information about the different subgroups of persons with PD and does not provide causal evidence of factors that may drive levodopa-resistant FoG in PD.

INTRODUCTION

The introduction is quite short and provides only a limited background of information regarding FoG in PD. There are significant bodies of literature investigating the motor response to levodopa in persons with PD and FoG that could help the reader to understand the direction and hypotheses of the study. As currently written, the introduction lacks focused hypotheses about what motor and non-motor features may be expected to differ among the groups of interest and why. Such hypotheses could be formed logically with a more thorough background of proposed mechanisms of both the motor response to levodopa and development of FoG in PD. What might we expect to differ between persons with and without levodopa-resistant FoG and why?

While I agree with the authors in that I am also not familiar with any studies that have directly compared persons with PD and levodopa-resistant FoG to those with levodopa-responsive FoG, there have been studies contrasting persons with PD and levodopa-resistant FoG against a more general population of persons with PD (e.g., Ferraye et al., Eur Neurol, 2013). These studies are relevant but currently omitted from the background.

METHODS

Given the stated goal of the study (identifying differences between persons with levodopa-resistant/levodopa-responsive FoG), it is not clear why a no-FOG group was included.

What was the reason for excluding frequent fallers? I understand that there could be safety concerns, but one would imagine that procedures could be put in place to minimize fall risk. It seems like omitting frequent fallers would exclude many potential participants particularly within the ON-FOG group.

Did all participants take their medications at the same time relative to the start time of their participation in the study (e.g., one hour prior to participation)?

It would be helpful to define the gait parameters “foot-strike-length” and “ambulation time”.

Line 84 – “were enrolled” is repeated.

Line 141 – what is meant by “low, middle, and high values of motor UPDRS”?

RESULTS

Several of the results sections are reported very descriptively without statistics to support claims made. This makes interpretation of the results challenging because it is not clear which comparisons resulted in statistically significant differences vs. which comparisons are being made qualitatively. Please include results from your statistical analyses throughout this section.

Please revise the headings in Figure 3 to read “affected” rather than “effected”.

DISCUSSION

In line with the comment above regarding the results, it is difficult to comment on the discussion given that it is not clear which results were statistically significant.

I found the contextualization of the findings within previously proposed models of FoG to be challenging to follow. It was not clear to me how comparisons being made between two populations that both experience FoG (levodopa-responsive FoG and levodopa-resistant FoG) provide evidence for or against models of FoG development; it would seem that comparisons between populations with and without FoG would be more insightful in testing these models.

Reviewer #2: INTRODUCTION:

• Incorrect use of the word mechanistic line 71

• Line 72, are non-motor features not objective? Clarify

• Should discuss prior work on levodopa induced FOG, and pseudo-ON FOG, as well as work on cognitive differences between pharmacologic subtypes.

• Provide reference for the assertion that FOG is initially dopa-responsive then becomes unresponsive

METHODOLOGY

• Clarify criteria for non-freezers. Did they have to have a 0 on item one of the fogq?

• Why was the nfogq not used? If not, this is a limitation of the study and should be mentioned. The original version had several limitations which is why the new version was introduced

• classification of ON FOG is unclear and confusing as it is written, please rewrite sentence beginning with “freezers…” in line 93

o it appears you are including levodopa induced FOG as ON FOG

o discussion regarding multiple classifications of dopa response is critical here. Some groups will define ON FOG as dopa induced FOG, others as any FOG in the ON state, and others as FOG that does not respond to dopamine, it looks like all of these were included in this group making it quite heterogenous

o OFF FOG group may also classified based on its dopa response

• There was no evaluation of objective gait parameters in the OFF state, major limitation

• It looks like there was no adjustment for multiple comparisons

RESULTS

• Please indicate on table 1, which differences were statistically significant

• Line 184, clarify what evidence there was of any differences

• Table 2 needs to be clarified, label specifically what factors are statistically significantly different, not just state that CI’s that don’t include zero are significant, there is no legend to explain what the bolded entries represent

• Cannot report “statistically significant” differences if did not correct for multiple comparisons

• Should not report non statistically significant differences even if there were trends that were not significant due to variance

DISCUSSION

• “In FOG participants with approximately the same disease duration, and age at onset, ON-FOG participants had greater cognitive dysfunction, greater motor disease severity, and lower quality of life than OFF-FOG participants. “ All of this is well known and previously reported. Authors should acknowledge this is well known, and reference prior work in the initial paragraph.

• I see no value in reported and much less discussing non significant trends.

• Sentence starting in line 278 needs a reference

• It does not follow that based on the fact that there are inter episodic gait changes in FOG there are four models of FOG. There are many other models than those proposed in 2013.

• I don’t see how the discussion of these 4 models is helpful in explaining the findings of the work presented here.

• I suggest only statistically significant findings between groups that survive multiple comparisons be reported and discussed.

• Specifically there is no discussion of the findings of figure 1 which imply that the changes that are seen, if they survive multiple comparisons are more likely due to overall motor decline that FOG itelf

Reviewer #3: Manuscript: Levodopa ON-state Freezing of Gait: Defining Gait and Non-motor Phenotype

This is an excellent contribution to freezing of gait in Parkinson’s disease, both in the OFF and ON states. The additional data on non-motor features distinguishing these subtypes further enhances this study.

Overall significance of the paper:

This study demonstrated that ON freezers had greater disease severity, cognitive deficits, and lower quality of life as compared to OFF freezers with similar disease duration.

The authors acknowledged a limitation in not exploring the freezing episodes themselves in detail, in addition to the spatiotemporal indices pre- and post- freezing. This would be a useful next step, as it may yield further distinguishing features between these 2 subgroups.

In addition, a future study utilizing 3D motion capture (in this cohort or perhaps collaborating with another institute with such data) would provide further data points, with particular attention to 3D kinematics of arms (i.e. arm-swing), leg swing, and heel/foot strike.

Other:

Page 4: Line 84 – Duplicate “were enrolled.” Please delete one.

6. PLOS authors have the option to publish the peer review history of their article (what does this mean?). If published, this will include your full peer review and any attached files.

Reviewer #1: No

Reviewer #2: No

Reviewer #3: No

---

## [Author Response · Author response to Decision Letter 0]

15 Sep 2021

Please see attached document "response to reviewers"

---

## [Decision Letter · Decision Letter 1]

4 Jan 2022

PONE-D-21-16241R1Levodopa ONOFF-state Freezing of Gait: Defining the Gait and Non-motor PhenotypePLOS ONE

Dear Dr. Virmani,

Thank you for submitting your manuscript to PLOS ONE. After careful consideration, we feel that it has merit but does not fully meet PLOS ONE’s publication criteria as it currently stands. Therefore, we invite you to submit a revised version of the manuscript that addresses the points raised during the review process.

As you can see, the reviewers are still divided about your article. I urge you to respond to all comments carefully. And especially adjust the text about aspects of the statistical procedures and stick only to what has been tested and evaluated quantitatively. For ease of review, please respond to ALL comments, indicating in the response letter where changes have been made (lines and pages) in the manuscript and even copying and pasting these parts of the manuscript into the response letter.

We look forward to receiving your revised manuscript.

Kind regards,

Leonardo A. Peyré-Tartaruga, Ph.D.

Academic Editor

PLOS ONE

Reviewers' comments:

Reviewer's Responses to Questions

**Comments to the Author**

1. If the authors have adequately addressed your comments raised in a previous round of review and you feel that this manuscript is now acceptable for publication, you may indicate that here to bypass the “Comments to the Author” section, enter your conflict of interest statement in the “Confidential to Editor” section, and submit your "Accept" recommendation.

Reviewer #1: (No Response)

Reviewer #2: All comments have been addressed

Reviewer #3: All comments have been addressed

2. Is the manuscript technically sound, and do the data support the conclusions?

Reviewer #1: Yes

Reviewer #2: Yes

Reviewer #3: Yes

3. Has the statistical analysis been performed appropriately and rigorously? 

Reviewer #1: No

Reviewer #2: Yes

Reviewer #3: Yes

4. Have the authors made all data underlying the findings in their manuscript fully available?

Reviewer #1: Yes

Reviewer #2: Yes

Reviewer #3: Yes

5. Is the manuscript presented in an intelligible fashion and written in standard English?

Reviewer #1: Yes

Reviewer #2: Yes

Reviewer #3: Yes

6. Review Comments to the Author

Reviewer #1: The authors have partially addressed the prior comments. The statistical reporting remains an issue. If the authors choose not to perform statistical analyses on certain aspects of the data, they should refrain from interpreting the data; that is, if demographic data are included to provide purely descriptive information about each group, the authors should not make statements such as "were not notably dissimilar" and "clearly the highest" without some type of quantitative statistical analysis to substantiate these claims. Certainly, the reader can see the data and ascertain for themselves which numbers are larger than others without additional need for interpretation if there are no statistical results to report.

Table 3 provides indicators of statistical significance but does not provide specific test statistics or p-values (as are provided for the gait data in the S1 Table).

Reviewer #2: The authors present a well written report of a comparative study between PD patients with freezing exclusively in the OFF state and those that have FOG in the ON and in the OFF state. They collect data on multiple domains to replicate prior work, and focus on reporting of quantitative gait findings that may separate the groups. While I do agree that quantitative measurements of gait, outside the FOG episode is of value in FOG, conceptually, there is a flaw in the premise that there would be phenomenological differences in based on dopa-response. Largely, this study fails to find significant behavioral differences in the recorded spatiotemporal parameters between OFF and ONOFF FOG, which is the main premise of the study, and the authors attribute to variability. The differences that are observed in cognition, quality of life, etc., can largely be explained by a greater disease severity. Moreover, these findings have been previously reported, particularly in cognition, with more thorough analyses, and this study did not add significantly to these findings. The discussion is well written and extensive, however, I do not see how the discussion of models of FOG is relevant to the findings of the study. I do agree on their depiction of the literature, the models, and dopa-response in FOG as a continuum, but again, I do not see how the data they present supports these assertions. There is a very transparent presentation of the data, and their analysis, and given the data presented, I believe their statistical approach, whereby FDR and effect sizes are presented, is justified.

Reviewer #3: I agree with Reviewer #1 regarding renaming the various FOG subgroups as it was difficult to follow the data at times without this added detail.

7. PLOS authors have the option to publish the peer review history of their article (what does this mean?). If published, this will include your full peer review and any attached files.

Reviewer #1: No

Reviewer #2: No

Reviewer #3: No

---

## [Author Response · Author response to Decision Letter 1]

4 Feb 2022

please see attached response to reviewers document.

---

## [Editor Report · Decision Letter 2]

26 Apr 2022

PONE-D-21-16241R2Levodopa ONOFF-state Freezing of Gait: Defining the Gait and Non-motor PhenotypePLOS ONE

Dear Dr. Virmani,

Thank you for submitting your manuscript to PLOS ONE. After careful consideration, we feel that it has merit but does not fully meet PLOS ONE’s publication criteria as it currently stands. Therefore, we invite you to submit a revised version of the manuscript that addresses the points raised during the review process.

The authors performed a hard work replying all questions raised by the reviewers. I have just a couple of minor comments. Firstly, I miss a discussion in the discussion (particularly in lines 260-280) benchmarking the results with high level o evidence on gait biomechanics of PD (PMID: 33436993). Secondly, not compulsory, but as a suggestion, try to reply these questions: how do your paper may help to health professionals to prescribe exercise for PD from your findings? (PMID: 34803726 ). Are there some 'bridges' between akinetic-rigid and hyperkinetic PD and your findings? PMID: 33517030, PMID: 34803726. Please submit your revised manuscript by Jun 10 2022 11:59PM. If you will need more time than this to complete your revisions, please reply to this message or contact the journal office at plosone@plos.org. Please include the following items when submitting your revised manuscript:A rebuttal letter that responds to each point raised by the academic editor and reviewer(s). You should upload this letter as a separate file labeled 'Response to Reviewers'.A marked-up copy of your manuscript that highlights changes made to the original version. You should upload this as a separate file labeled 'Revised Manuscript with Track Changes'.An unmarked version of your revised paper without tracked changes. You should upload this as a separate file labeled 'Manuscript'.If applicable, we recommend that you deposit your laboratory protocols in protocols.io to enhance the reproducibility of your results. Protocols.io assigns your protocol its own identifier (DOI) so that it can be cited independently in the future. For instructions see: https://journals.plos.org/plosone/s/submission-guidelines#loc-laboratory-protocols. Additionally, PLOS ONE offers an option for publishing peer-reviewed Lab Protocol articles, which describe protocols hosted on protocols.io. Read more information on sharing protocols at https://plos.org/protocols?utm_medium=editorial-email&utm_source=authorletters&utm_campaign=protocols.

We look forward to receiving your revised manuscript.

Kind regards,

Leonardo A. Peyré-Tartaruga, Ph.D.

Academic Editor

PLOS ONE
---

## [Author Response · Author response to Decision Letter 2]

27 Apr 2022

Please see attached file with response to reviewers.

---

## [Editor Report · Decision Letter 3]

18 May 2022

Levodopa ONOFF-state Freezing of Gait: Defining the Gait and Non-motor Phenotype

PONE-D-21-16241R3

Dear Dr. Virmani,

We’re pleased to inform you that your manuscript has been judged scientifically suitable for publication and will be formally accepted for publication once it meets all outstanding technical requirements.

Kind regards,

Leonardo A. Peyré-Tartaruga, Ph.D.

Academic Editor

PLOS ONE

---

## [Editor Report · Acceptance letter]

23 May 2022

PONE-D-21-16241R3 

Levodopa ONOFF-state Freezing of Gait: Defining the Gait and Non-motor Phenotype 

Dear Dr. Virmani:

I'm pleased to inform you that your manuscript has been deemed suitable for publication in PLOS ONE. Congratulations! Your manuscript is now with our production department. 

Kind regards, 

on behalf of

Professor Leonardo A. Peyré-Tartaruga 

Academic Editor

PLOS ONE